# Guidance for Interactive Visual Analysis in Multivariate Time Series Preprocessing

**DOI:** 10.3390/s25185617

**Published:** 2025-09-09

**Authors:** Flor de Luz Palomino Valdivia, Herwin Alayn Huillcen Baca

**Affiliations:** Faculty of Engineering, Academic Department of Engineering and Information Technology, Jose Maria Arguedas National University, Andahuaylas 03701, Peru; hhuillcen@unajma.edu.pe

**Keywords:** guidance, interactive visual analysis, multivariate time series, preprocessing, recommendation, explainability

## Abstract

Multivariate time series analysis presents significant challenges due to its dynamism, heterogeneity, and scalability. Given this, preprocessing is considered a crucial step to ensure analytical quality. However, this phase falls solely on the user without system support, resulting in wasted time, subjective decision-making, and cognitive overload, and is prone to errors that affect the quality of the results. This situation reflects the lack of interactive visual analysis approaches that effectively integrate preprocessing with guidance mechanisms. The main objective of this work was to design and develop a guidance system for interactive visual analysis in multivariate time series preprocessing, allowing users to understand, evaluate, and adapt their decisions in this critical phase of the analytical workflow. To this end, we propose a new guide-based approach that incorporates recommendations, explainability, and interactive visualization. This approach is embodied in the GUIAVisWeb tool, which organizes a workflow through tasks, subtasks, and preprocessing algorithms; recommends appropriate components through consensus validation and predictive evaluation; and explains the justification for each recommendation through visual representations. The proposal was evaluated in two dimensions: (i) quality of the guidance, with an average score of 6.19 on the Likert scale (1–7), and (ii) explainability of the algorithm recommendations, with an average score of 5.56 on the Likert scale (1–6). In addition, a case study was developed with air quality data that demonstrated the functionality of the tool and its ability to support more informed, transparent, and effective preprocessing decisions.

## 1. Introduction

Multivariate time series analysis represents an increasing challenge in multiple application areas due to temporal dynamism (continuous variations over time), heterogeneity (data from diverse sources and geographic locations), and scalability (ability to handle large volumes of data). A thorough preprocessing process is essential before implementing modeling or prediction techniques. However, this procedure is highly dependent on context and user knowledge, making it a crucial and often difficult phase to implement [1].

In this context, visual analytics (VA) [2,3] has proven to be an effective strategy for facilitating the exploration and understanding of time series. However, its application has focused on the analysis phase, neglecting preprocessing, a critical step that directly affects the quality of subsequent analysis [4,5]. This lack of integration means that this preprocessing step is delegated to the user, who, at their discretion, must make complex technical decisions without explicit support from the system. This situation generates cognitive load, knowledge gaps, subjective and overloaded interpretations, high time consumption, and, in many cases, frustration due to not achieving the expected results [6].

Approaches are emerging that begin to identify the importance of preprocessing in interactive visual analysis. For example, PrAVA [4] proposes including this stage as an essential component of the AV workflow, highlighting the importance of supporting the user not only in the analysis but also in understanding and evaluating the effects of the applied preprocessing. However, there are still gaps in its effective incorporation as an assisted stage within interactive analysis systems [1].

A growing strategy to simplify these tasks is the implementation of guidance mechanisms. Guidance helps users understand data, address analytical challenges, and make informed decisions [2,6,7,8].

Therefore, we propose a new approach, “Guidance for Interactive Visual Analysis in Multivariate Time Series Preprocessing”, through the automated generation of recommendations for tasks, subtasks, and preprocessing algorithms. For task and subtask prioritization, a comparative evaluation of algorithms is implemented along with a majority consensus-based validation mechanism. Furthermore, the selection of optimal algorithms is based on the execution of predictive models, contrasted with quantitative validation metrics, which allows for the identification of the most suitable ones for the analysis.

Guidance in visual analytics systems is essential; however, a lack of precise explanations can hinder its effectiveness. While explainability is addressed in Artificial Intelligence, it still lacks sufficient attention in guidance. Therefore, our proposal includes explanations of the reasons why algorithms are recommended to reinforce user confidence and provide clarity for the analysis process [8,9,10,11,12,13].

Our proposed guidance approach relies on two main pillars: (1) a conceptual model of interactive guidance, with four interconnected modules and a guided graph, and (2) an interactive visual tool that implements this model in an adaptable analysis environment.

To evaluate the proposal, we first evaluated the explainability model of the recommended algorithms. The results indicate a robust system that fosters trust, and the explanations meet expectations for clarity, timeliness, relevance, and cognitive ease. Second, we evaluated the guides, whose results indicate positive feedback, highlighting their expressiveness, relevance, visibility, clarity, adaptability, and timeliness. Third, we conducted a case study of air quality using the Madrid time series, demonstrating the functionality of the GUIVisWeb tool.

### 1.1. Problem

The problem lies in the absence of interactive visual analysis approaches that integrate multivariate time series preprocessing with guidance mechanisms. Without this support, the user faces a high cognitive load and must make complex decisions individually, which increases the risk of errors and compromises the quality and consistency of the results.

### 1.2. Objectives

The objective was to design and develop a new guidance approach that integrates recommendations, explainability, and interactive visual analysis to assist users in the preprocessing of multivariate time series, supporting decision-making, reducing cognitive load, and improving the quality and reliability of the analytical workflow.

We designed a modular architecture that incorporates guidance levels, algorithm recommendations, and explainability mechanisms to bridge the knowledge gap and support decision-making in multivariate time series preprocessing.We designed a systematic workflow by identifying and organizing tasks, subtasks, and algorithms for multivariate time series preprocessing.We developed and implemented an interactive visual analysis tool that embodies the proposed architecture and assists users in preprocessing multivariate time series.We evaluated the proposed approach through case studies and user studies, measuring criteria of usability, explainability, and quality of results.

### 1.3. Contributions

The following contributions are significant to the state of the art:The preprocessing phase is incorporated as part of the interactive visual analysis process for multivariate time series.A proposal of a guidance system for interactive visual analytics, focused explicitly on preprocessing multivariate time series, is given.An introduction of automatic recommendations for tasks, subtasks, and algorithms as a key element of the guidance system is provided.The integration of visual explainability within the guidance process is achieved, extending the concept of explainability (widely used in AI/XAI) to the novel notion of explainable guidance (XG).This study contributes to the scientific community through a set of reproducible strategies for integrating guidance into multivariate time series analysis, including workflow templates and validation criteria.

### 1.4. Paper Structure

The remainder of this paper is organized as follows. Section 2 presents related work on visual analysis, guides, and multivariate time series preprocessing. Section 3 details the proposed approach: architecture and its respective modules. Section 4 describes experiments with a case study and the results. Section 5 reports discussions on future work. Finally, Section 6 presents the conclusions.

## 2. Related Work

In this section, we present the papers we used as a reference for developing our proposal. We explore works on visual analysis with guides, preprocessing, null value imputation techniques, visual interfaces, explainability, and guide evaluation.

### 2.1. Works on Visual Analysis with Guides

Ceneda et al. [6] propose a theoretical model that characterizes guidance in visual analytics according to knowledge gaps, inputs, and outputs. Guidance fluctuates in degrees of orientation, direction, and prescription. Our proposal addresses guide grades, which are essential in any guiding work.

The study by Ceneda et al. [14] focuses on the visual exploration of cyclic patterns in univariate time series using spiral visualization, with precise suggestions of visual parameters and clues, which increases user satisfaction and confidence in the identified patterns. We use it as a reference for cycle detection in multivariate time series, and we have published a paper on univariate and multivariate analysis [15].

On the other hand, Luboschik et al. [16] propose directing multiscale data analysis towards regions with varying data behavior. While they answer the question, “What data is important?”, they explore large datasets and highlight specific regions. Our approach addresses the question “How can we analyze them?” through data analysis; we also use visual signals to make recommendations.

Similarly, May et al. [17] present an approach that uses visual cues for visual navigation through large abstract graphs. We adopt their concept to use the graph; however, in contrast to the static graph, our proposal employs a dynamic graph with a hierarchical structure.

The research by Streit et al. [3] introduces a three-stage model-based design process for interactive visual analysis of heterogeneous datasets. Unlike their proposal, which bases its visualization on the use of predefined static models and established paths within a specific domain, our proposal uses a dynamic graph with an adaptive hierarchical structure that does not depend on predefined domain models. Instead, it generates recommendations based on quantitative validation, enabling dynamic integration between technical and cognitive guidance, a feature that static approaches fail to achieve.

In the same way, Gladisch et al. [18] represent goal navigation in a hierarchical graph. While navigation recommendations answers the question “Where is the relevant information?” through visual cues for the user to pay attention to the suggested point, the graph proposed in our research answers the question “How to analyze it?”. It addresses the lack of structure in the workflow, allowing for the execution of algorithms on each defined task.

The study by Collins et al. [7] provides general guidance that addresses knowledge gaps, with a focus on decision-making processes. It reduces errors, mitigates biases, and alleviates cognitive load. Our approach incorporates guidance in both interaction tasks and analytical processes. We use quantitative metrics and majority consensus mechanisms for algorithm recommendation, which enables us to incorporate both cognitive and technical aspects into the analysis of multivariate time series.

In the work of Han et al. [19], the focus is on the wayfinding problem as a decision-making problem, using decision-making theory and models. They propose a three-stage method for users to compare and make choices: identifying decision points, deriving and evaluating alternatives, and visualizing the resulting alternatives. Our approach consolidates algorithm selection and validation into a structured workflow, overcoming the rigidity of traditional systems by using metrics and dynamic context.

Perez et al. [20] propose a typology of guide tasks for the system, considering why, how, what, and when. They introduce perspective change, illustrating how different degrees of guidance affect user search performance. This tool aids in designing and analyzing guided visual analysis systems, ensuring practical and understandable interactions.

Table 1 summarizes the reviewed research works. Although some references are not recent, they provided a solid foundation for our proposal. The table shows a predominance of theoretical approaches (frameworks, conceptual models, typologies) over practical implementations, which, when present, are usually concrete. The studies address knowledge gaps from different perspectives, focusing on aspects such as algorithm application, interface design, or system integration. Guidance levels are often combined, and recent works move toward more integrated perspectives that incorporate cognitive, technical, and decision-making dimensions, showing an evolution from conceptual models to practical applications.

### 2.2. Works on Preprocessing

To understand the most common preprocessing methods (tasks), we conduct a review of the state of the art.

The authors agree that data preprocessing is essential since the data must be adapted for a modeling process because, if the data is inadequate, the model algorithms can fail during execution, affecting the efficiency and accuracy of the results [22,23]. Fan et al. [5] review the state of the art of the most common preprocessing tasks. The conceptual approach by Milani et al. [4] (PrAVA model) emphasizes the need to address preprocessing as one of the most critical phases in visual analytics (VA) processes since this approach does not incorporate preprocessing as an important phase of the process, considering that 80% of the effort is concentrated between preprocessing and obtaining results, which makes it a relevant activity, often considered implicit or secondary in conventional visual analytics (VA) models. This conceptual approach motivates researchers to consider preprocessing as an integral part of visual analysis. Our proposal integrates and develops preprocessing tasks within VA; evaluations were conducted based on the nine guidelines specified by Milani et al.

The review by Maharana et al. [24] discusses the latest advancements in preprocessing and data augmentation techniques in machine learning, focusing on transforming data with noise, missing values, and inconsistencies; cleaning; normalization; noise; and dimensionality reduction. Likewise, Ccetin et al. [23] provide a review of data preprocessing methods for data analysis, achieving superior results by combining various methods. Like Palomino et al. [21], their proposal presents preprocessing methods that are applied before performing time series analysis.

The study by Mallikharjuna et al. [25] aids in various data preprocessing techniques, including missing data handling, categorical feature encoding, discretization, outlier detection, and feature scaling, to build effective predictive models.

Table 2 shows a summary of various studies based on tasks of the preprocessing phase, which allowed us to select the most common ones for our proposal. There is a consensus on the tasks used: (1) Transformation, (3) Normalization, (4) Missing data imputation, (5) Noise, (6) Outliers, and (7) Dimensionality reduction.

### 2.3. Works on Null Value Imputation Techniques

Emmanuel et al. [26] review missing data imputation techniques, highlighting machine learning-based methods as the most effective. The reviewed techniques include (1) mean, median, and mode; (2) statistical methods like linear/multivariate regression, hot-deck imputation, and expectation–maximization (EM); and (3) machine learning approaches, such as k-nearest neighbor (kNN), random forest, support vector machines (SVMs), decision trees, clustering (k-means and hierarchical), and ensemble methods (bagging, boosting, stacking).

Hasan et al. [27] made a review of publications in which they agree with other authors in recognizing that KNN and random forest are the most used and with the best results, as well as regression and the classic statistical methods based on the mean and median.

Joel et al. [28] highlight missing values as a key challenge in real-world data, noting that the accuracy of machine learning models relies on data quality. The article reviews techniques for handling missing data, including mean, median, regression, KNN, and decision trees.

Similarly, Thomas et al. [29] examine the suggested techniques for data imputation, especially in the area of machine learning. They also consider the best techniques to be KNN, decision trees, and algorithms based on linear and additive regression, as well as the mean and median algorithms.

Based on the statistics from the aforementioned reviews, we compiled a list of the techniques (Table 3) and their frequency in Figure 1.

Table 3 shows that the most widely used technique isk-nearest neighbor (KNN). Traditional statistical methods such as mean and median show significant adoption, as do linear regression and random forests, while approaches such as polynomial regression and Bayesian PCA are not widely used.

### 2.4. Works on Explainability

Zhang et al. [9] implement GuidedStats, a computational notebook extension that facilitates the transition between graphical interfaces and code, enabling users to perform statistical analysis through guided, interactive workflows. Ha et al. [10] found that while visual explanations are helpful in complex tasks, they do not significantly impact users’ trust in AI suggestions. This suggests that explainability strategies may need to be tailored to specific tasks and cognitive load. On the other hand, Sperrle et al. [13] provide a comprehensive overview of evaluations in human-centered machine learning, focusing on trust, interpretability, and explainability, relating to model interpretability and transparency. The proposal by Musle et al. [8] presents a model for developing explainability in guided visual analytics systems, aiming to increase user trust. It incorporates concepts from explainable artificial intelligence (XAI), guidelines for designing guides in visual analytics, and trust dynamics between the user and the system. Integrating explainability into guides—a new concept—is a valuable contribution.

Explainability has been widely discussed in artificial intelligence (XAI) and visual analytics, but not in explainable guidance (XG) with guides. Therefore, we turned to AI research to apply it to the recommendation algorithms in our proposal so that users understand the reason behind the system’s suggestions (enhancing transparency and trust).

### 2.5. Works on Guidance and Explainability Evaluation

To evaluate our explainability model applied to the preprocessing algorithms recommended by the system, we rely on Musleh et al. [8], who propose a model for designing explainability strategies in guided visual analytics systems, based on the Dynamics of User Trust and Explainability Features. These dynamics help identify explainability features by considering the dynamic relationship between the user and the system, the development of user trust in the system, and guidance suggestions. These features have four main requirements for improving guidance effectiveness: not overloading the cognitive process, following guidance cues, integrating explainability cues into the process when necessary, and avoiding uncertainty in the understanding process.

The dynamics of user trust (stages of trust relationships and user expectations) and characteristics of explainability are inputs for the context (Who is the user interpreting the explanation? Which guidance requires explanations? Why does the user need an explanation?), and this, in turn, makes up the structure (What explanation does the user need? How does the system present the explanation?) and setting (When does the system present an explanation? Where does the system present the explanation?).

A series of components specified in Musleh et al. [8]’s model were taken into account to evaluate our proposal.

To evaluate the guidance system of our proposa, we rely on the work of Ceneda et al. [30], who propose a methodology for evaluating visual analysis work with guides, in which they identify eight quality criteria: Flexible (dynamic adjustment of the degree and type of support; identifying an adequate degree and the capacity of the system to modify the amount and type of support, essential for each user’s task), Adaptive (the content of the suggestions is adapted through analytical information, preferences, habits, and interactions with previous suggestions, which can be used to generate customized content for the user), Visible (the state of the guide is visible to the user through adequate visual feedback and in a reasonable time), Controllable (the user must be able to adjust and control the parameters that affect the production of the guide), Explainable (the way in which the user communicates and understands the guide; each action and suggestion of the system must be adequately explained to avoid confusion), Expressive (guide suggestions can be provided and communicated using language appropriate to the user, avoiding confusion and ambiguity), Timely (the guide is provided at the right time), and Relevant (the guide can guide the user towards relevant analysis results, avoiding errors in the process). The guide should have a goal and be accomplished with basic user support tasks. The system should support the exploration of alternatives when several actions are possible, depending on the analysis context.

We evaluated our proposed preprocessing method based on the nine guidelines outlined by Milani et al. [4]: G1: Integration with the most commonly used data analysis tools. G2: Ability to work with large-volume data scenarios. G3: Generation of informative summaries of preprocessing activities. G4: Use of data mining to support preprocessing activities. G5: Statistical methods to describe data and support preprocessing. G6: Compare data before and after transformations and their impacts. G7: Recommender systems to propose visualizations. G8: Automatic generation of initial visualizations or templates. G9: Visual interaction for flexible data exploration.

## 3. Proposal

According to the state of the art, and in response to the challenges inherent in proposing guidance for interactive visual analysis in the preprocessing of multivariate time series, a comprehensive architecture was developed that addresses both technical and cognitive aspects, covers the knowledge gap [6,14,16,17], and facilitates appropriate decision-making [7,19,31].

### 3.1. Proposal Architecture

The overall proposal consists of two main components: on the one hand, guidance for interactive visual analysis in the preprocessing, which conceptualizes functions based on four interconnected modules and a guided graph, and, on the other hand, the tool for guided visual analysis, which allows the implementation of these guides in a practical environment (Figure 2).

### 3.2. Component of Guidance for Interactive Visual Analysis in the Preprocessing

This component is structured into four modules and a guided graph.

#### 3.2.1. Configuration Module

We structure the user–system interaction around four key elements: tasks, subtasks, specific algorithms, and visual interfaces.

Definition of tasks and subtasksWe defined primary tasks (Table 4) and subtasks (Table 5), according to the state-of-the-art review conducted in Section 2.2. They included tasks for trend, seasonality, and cyclicality analysis.Definition of algorithmsBased on prior experience and analysis of the state of the art, a selection of algorithms was made for each task/subtask. The focus was on the publications on algorithm review:
-Identification of null value imputation techniques: The proposal is based on using techniques to impute data, rather than eliminate it [26,27,28,29].Based on the state-of-the-art analysis conducted in Section 2.3, Table 3, we selected all proposals with multiple references of two or more. Clustering can distort sequential patterns such as trends, seasonality, or temporal interrelationships between variables. On the other hand, SVMs are not designed to model temporal dependencies or relationships between variables, and, for multivariate regression tasks, they require complex architecture. For all these reasons, we ruled out SVM and clustering.There is a little-referenced technique known as Legendre Polynomials. This technique yielded promising results in tests with our time series.-Techniques for treating outliers: A hybrid approach is proposed based on the combination of two robust techniques: the Z-score [32,33] and the Interquartile Range (IQR) [34]; this approach is particularly beneficial for large volumes of data.-Techniques for normalization: Based on the most referenced scientific articles on normalization techniques, the most robust and widely used techniques for multivariate time series were identified [35,36]. These techniques allow data preparation when the variables present different magnitudes or variations, facilitating a fair comparison between them without requiring a nonlinear transformation or altering the temporal pattern of the data.-Techniques for transformation: The most frequently used data transformation techniques in time series processing were identified [35,37]. These techniques allow for variance reduction, trend correction, smoothing seasonal patterns, or stabilizing the series for subsequent analysis.-Techniques for dimensionality reduction: According to the state of the art, it was proposed to use the two most representative techniques for dimensionality reduction: principal component analysis (PCA) and factor analysis (FA). In PCA, data distribution is not a requirement, and it is widely used for data whose variables are highly correlated, making it computationally efficient. FA is beneficial when one wants to find latent correlations [38,39].
Finally, a summary of all the algorithms used for the tasks/subtasks of the workflow are presented in Table 6.Definition of visual interfacesThe design of visual interfaces was not oriented toward graphic sophistication but rather toward clarity and expressiveness. For the main workflow visualization, a dynamic graph with a hierarchical structure was adopted as it best expressed structured navigation.The selection of visual interfaces is shown in Table 7.

#### 3.2.2. Workflow Module

An integrated workflow was established that organized and related the elements defined in the configuration module in a logical sequence, ensuring methodological consistency throughout the analysis [3,21]. This approach has several advantages: (1) Systematicity, since each technical decision (selection of algorithms, parameters, or visualizations) is organized hierarchically, where the results of one task/subtask serves as input for the next. (2) Modularity, since the design allows for replacing or expanding specific components without modifying the overall system architecture. (3) Reproducibility, since by standardizing each stage of the analysis—from preprocessing to the interpretation of temporal behavior—the model facilitates the repetition of the analysis in different contexts, guaranteeing traceability and consistency, even in complex and multivariate scenarios (Figure 3).

The workflow organization follows a hierarchical structure, starting with the group (G), which groups tasks (T). Each task is divided into subtasks (S) that describe specific actions, each with assigned algorithms (A) (Table 8).

#### 3.2.3. Recommendation Module

Our recommendation strategy involves a global execution process of the algorithms prior to analysis, the results of which allow us to more accurately recommend the tasks, subtasks, and algorithms that best align with the nature of the data [22,40].

Furthermore, stability is a crucial characteristic when choosing algorithms. However, there are many alternatives; the most widely used algorithms tend to have lower error rates due to their extensive use and the progressive improvements made in previous studies. Furthermore, their popularity is generally supported by detailed documentation and active support in the scientific and technical community [41].

When implementing the algorithms, we used standard thresholds to identify extreme values, considering the highest and lowest values within the data distribution. These thresholds are frequently used in the literature and practical applications as they mitigate the occurrence of false positives and false negatives depending on the analysis context [42].

Finally, the system not only suggests what to do but also communicates the reason for an algorithm’s recommendation.

Recommendation of tasks and subtasks.
-Multiple validation algorithms are run (there may be 2, 3, or more).-Each algorithm returns a Boolean value TRUE or FALSE, depending on specific conditions that justify the need to implement a task or subtask.-If at least 50% of the algorithms return TRUE, the system suggests implementing the task or subtask.-If the majority does not exceed 50%, the task or subtask is not recommended.


Below are the details of the recommendation for each task or subtask; there is no single criterion as each task has its particularity:
(A)Cleaning: The cleaning task recommendation is based on a combination of different statistical techniques to identify noise and outliers:
-Calculates noise for each variable based on the standard deviation if the values are outside [μ−5·σ,μ+5·σ].-Calculates noise for each variable based on the coefficient of variation according to the following condition: $\left(CV=\frac{\theta }{\mu }\;2\right)$.-Detects outliers for each variable, for which it calculates the IQR (IQR = Q3 − Q1); then, values outside the limits [Q1 − 1.5.IQR, Q3 + 1.5.IQR] are considered outliers.-Detects outliers for each variable with a Z_Score, where those greater than or equal to 3.5 are considered outliers.-Applies the Grubbs test to detect an extreme value (minimum or maximum) for each variable, where α=(0.01).-Each test returns a True or False value.-Tallies the tests. If at least 50% of the tests detect problems in the data (returning True), a cleanup task is recommended.
(B)Outliers: Determines whether there are outliers in the data by implementing three primary methods: Z-score, IQR (Interquartile Range), and Grubbs Test.
-Calculates the Z-score for each variable (item iterates by column), using the default threshold of 3.5. Outliers greater than or equal to the threshold value are considered outliers. The result is the number of outliers and a Boolean value of True, indicating the existence of outliers.-Calculates the IQR for each variable (IQR = Q3 − Q1). It calculates the IQR limits, [Q1 − 1.5.IQR, Q3 + 1.5.IQR] and values outside the limits are considered outliers. The result is returned as the quantity and the Boolean value.-The Grubbs Test detects an extreme outlier, either the minimum or maximum per column (α=0.01). The result is returned as a Boolean value.-The methods used for statistical tests include standard deviation to detect noise and coefficient of variation to identify variability.-Additionally, noise detection (extreme peaks or high variability) is performed. For each variable, it calculates the mean (μ) and standard deviation of the series (σ) and then identifies outliers [μ−α·σ,μ+α·σ]. If there are outliers, it marks the column as noisy with the value True.-Finally, it counts the Boolean values equal to True; if the result is ≥50%, it concludes that outliers have been detected, so the task is recommended.
(C)Normalization: This is applied based on two main criteria: seasonality detection and distribution evaluation.
-Seasonality detection: The presence of seasonal patterns that may influence the distribution of the series is assessed. To do this, the MinMaxScaler is first applied to avoid scaling biases. A seasonal decomposition is then performed to separate the series into its trend, seasonality, and residual components. The variances of the seasonal and residual components are then compared. Suppose that the variance of the seasonal component is greater in at least one variable. In that case, it is concluded that seasonality impacts the distribution of the series, so a normalization process is recommended.-Distribution evaluation: This determines whether the data follow a known distribution (normal or log-normal) after scaling. It then scales the data using MinMaxScaler, calculates the standard deviation for each variable, and compares it with the standard deviation expected from a normal distribution. If the difference is slight, the series follows a normal distribution. Otherwise, it then attempts to fit a log-normal distribution to verify whether it belongs to a family of known distributions. If the series is not regular, it is marked as true.-Final Decision: Combines the test results to decide whether normalization is recommended, running the seasonality and distribution detection functions in parallel. From the results, if 50% or more of the tests return a value equal to true, this indicates the presence of seasonality or atypical data; therefore, normalization is recommended. Otherwise, it is not necessary. This activates all four available scaling types, leaving the final choice up to the user.-If no additional data quality issues are detected, but the series scales contain both negative and positive values, only MinMax normalization is enabled, as this option could be helpful to the user after analyzing the signal.
(D)Dimensionality reduction: This is carried out as follows.
-The correlation between variables is assessed using different methods (Pearson, Spearman, and Kendall) to measure the correlation between the scaled variables. If many variables are found to be highly correlated (above a defined threshold), this indicates that redundancies exist in the data that could be reduced.-Multicollinearity is detected: The Variance Inflation Factor (VIF) is calculated for each variable. If some variables are found to have a high VIF (greater than 10), this suggests the existence of high multicollinearity, which supports the need to reduce dimensionality.-Dimensionality reduction tests are performed, including principal component analysis (PCA), which evaluates the cumulative variance of the components. If the cumulative variance does not reach a threshold (0.8) for all components, dimensionality reduction is possible. Factor analysis (FA) is also performed, where the magnitude of the factor loadings is analyzed and variables with low values (below a threshold of 0.4) are identified for potential elimination.-If the majority of these tests (at least 50%) indicate redundancies or high correlation, it is concluded that dimensionality reduction is necessary.
(E)Transformation: The recommendation is based on the assessment of stationarity and persistence. To verify whether the time series is non-stationary, non-stationarity tests are applied, specifically the Augmented Dickey–Fuller (ADF) and Kwiatkowski–Phillips–Schmidt–Shin (KPSS) statistical tests. If either of these tests indicates non-stationarity, the series is considered a possible candidate for transformation.Additionally, to assess persistence, the Hurst exponent (H) is calculated, which measures whether the series exhibits persistent or anti-persistent behavior. If H>0.5, this means that the series exhibits a long-term trend and may require transformation to improve its behavior. This procedure enables us to determine whether a time series requires a transformation to become stationary or enhance its behavior before applying other analyses.


Recommendation of algorithms for null values.The defined algorithms (Table 6) are executed and evaluated using the Weighted Mean Absolute Percentage Error (WMAPE) metric to identify the most appropriate algorithm and prioritize the recommendation for execution (Figure A1 in Appendix A). The WMAPE metric is used because it is more robust compared to other existing metrics [43,44]. The best model is selected using the following criteria:$$wmape=\left\{\begin{array}{cc} \mathrm{good} & \mathrm{if}\;0.1\leq wmape\leq 0.2\\ \mathrm{acceptable} & \mathrm{if}\;0.2<wmape\leq 0.5\\ \mathrm{poor} & \mathrm{if}\;wmape>0.5 \end{array}\right.$$The null filling procedure uses both non-predictive and predictive models. In all cases, the dataset is split into three parts: training, validation, and test datasets. For predictive models, the model is trained on the training set, and the data is predicted using the validation set. The metric is obtained by evaluating the predicted data from the validation set against its actual values. The model with the best weighting is then used to predict the null dataset. A detailed explanation follows.(a)Given a series, an evaluation of the null values is performed (Table 9).(b)The null values are then separated by generating two datasets: one containing the complete data and another with the null data, ensuring that the original indexes are respected (Table 10).(c)Three datasets are then generated: Of the complete data, 80% is used to train the model, 20% is used for validation, and, from this, the metric is obtained by comparing the predicted data result with its actual data. The third null dataset is completed with the model that best weights according to the WMAPE metric (Table 11).(d)Finally, the entire series is assembled by completing the null data.For non-predictive models, the procedure is similar; however, instead of training the model with 80% of the whole dataset and validating it with the remaining 20%, the mean and median are calculated. These values are compared with the actual values of 20% of the whole dataset to obtain the metric evaluation.
-Implementation of algorithm explainability for null values.In order to provide transparency and facilitate understanding of the recommendation process, an explanatory visualization is implemented (a two-way matrix, relating algorithms and variables, and using the WMAPE metric as a key performance indicator), which allows the user to analyze why specific algorithms are suggested over others for handling null values (Figure 4).
(a)Y-axis (columns): Algorithms evaluated for imputation. X-axis (rows): Time series variables with null values. Matrix cells: WMAPE value obtained by each algorithm when imputing a given variable.(b)Cells highlighted in red indicate the lowest WMAPE per variable (best performance in terms of relative error). The visual highlighting makes it easy to identify which algorithms are most effective immediately. The system quantifies the number of times each algorithm achieved the best performance (minimum WMAPE).(c)Those algorithms with the highest number of red cells are prioritized and recommended.(d)In the dynamic hierarchical graph, prioritization is reflected with a more intense color in the nodes corresponding to the recommended algorithms (Figure A1 in Appendix A)).(e)When interacting with the visualization (by hovering over a node), additional contextual information is displayed: WMAPE definition and formula, recommended algorithm description, and WMAPE extreme values by variable.

Recommendation of algorithms for outlier detection.The selection of the appropriate algorithm is based on a sequential comparison strategy between two complementary approaches: Z-score (parametric) and IQR (non-parametric) (Figure A2 in Appendix B).
(a)The Z-score is applied and the detected outliers are replaced with the median of the series.(b)IQR is applied to the modified data to detect residual outliers. The system recommends the Z-score if there are no residuals; otherwise, it recommends the IQR.(c)Therefore, the Z-score is optimal when the data is normally distributed; if outliers persist after its application, IQR is recommended.
-Implementing explainability of algorithms for outliers.In order to make the algorithm selection transparent, a comparative visualization is implemented, based on a bar chart (Figure 5), where each bar represents an algorithm (Z-score or IQR) and the height indicates the percentage of outliers successfully treated by each method. In case of a tie (both methods treat 100% of the outliers), the use of the Z-score is privileged as the standard method, consistent with the assumption of normality and its computational efficiency [45].Recommendation of algorithms for dimensionality reduction.Various heuristic tests are applied to the data, and the results are then combined to determine whether dimensionality should be recommended (Figure A3 in Appendix C).
(a)The Kaiser Meyer Olkin (KMO) index is calculated on the scaled data. This index allows us to assess whether the dataset presents sufficient partial correlation to allow for the application of dimensionality reduction techniques.(b)If the KMO value is greater than or equal to 0.7, factor analysis (FA) is recommended, as the data are suitable for identifying latent factors.(c)If the KMO value is less than 0.7, principal component analysis (PCA) is used due to its robustness in scenarios with low correlation between variables.(d)The final recommendation is made by considering the consensus among the tests and selecting the method that best preserves the temporal structure of the series (trend and seasonality).
-Implementation of explainability of algorithms for dimensionality reduction.To explain the algorithm’s recommendation, the calculated value of the KMO index is visualized, accompanied by a correlation matrix that allows the user to explore the degree of correlation between variables visually (Figure 6).
(a)If KMO ≥ 0.7, it is recommended to use FA.(b)If KMO < 0.7, it is recommended to use PCA.(c)The explanation is complemented by textual information in the black and yellow background boxes.
Recommendation of normalization algorithms.The recommendation is based on a structured approach that seeks to preserve the temporal structure of multivariate series. This approach consists of two stages: (1) analysis of the need for normalization and (2) selection of the most appropriate scaling method.
(a)First, the data is checked for normalization by evaluating two key aspects: the presence of seasonality and its fit to a known distribution. If either of these two criteria is positive, the normalization recommendation process is activated.(b)To identify the most appropriate scaler, four versions of the same dataset are created using the MaxAbsScaler, MinMaxScaler, StandardScaler, and RobustScaler methods. Seasonal-Trend decomposition using LOESS (STL) is then applied to each scaled version and the original data, extracting the trend and seasonality components.(c)To evaluate which scaling method best preserves temporal structure, the correlation between the trend and seasonality components of the original version versus each scaled version is calculated. For each scaler, an average value of these correlations is obtained per variable, which represents its ability to preserve temporal dynamics.(d)The algorithm with the highest average correlation is considered the most suitable for the analyzed dataset.
-Implementing explainability of algorithms for normalization.The recommendation is accompanied by a bar chart, where each bar represents a scaling method and its average correlation value. This visualization makes it easy to compare the relative performance of the algorithms. The system reinforces this explanation with pop-up textual information boxes when interacting with each visual element, making the selection criteria easier to understand.Recommendation of transformation algorithms.The recommendation is accompanied by a bar chart in which each bar represents a scaling method and its average correlation value. This visualization allows for easy comparison of the algorithms’ relative performance. The system reinforces this explanation with pop-up textual information boxes when interacting with each visual element, facilitating understanding of the selection criteria.
(a)Augmented Dickey–Fuller (ADF): Detects the presence of non-stationarity.(b)KPSS: Evaluates stationarity in terms of levels.(c)Hurst Exponent (H): Measures the degree of persistence of the long-term series, which supports the transformation.(d)Each test returns a Boolean value. If at least 50% of the tests return TRUE, the system recommends applying a transformation.


#### 3.2.4. Guidance Level Module

Based on the characterization of the guidance presented by Ceneda et al. [6], we apply different degrees of guidance, adapting them to the user’s needs and ensuring they receive the appropriate level of support [30].

Orientation:The main objective is to build or maintain the user’s mental map. Our proposal implements the following:
-History of Actions: In the graph, the system uses the color blue on the nodes and edges to indicate the path of the executed actions, while stacked windows are displayed on the sides with the record of completed tasks.-Assigning views: The system implements statistical functions that allow for tracking changes in data in the preprocessing stage and analyzing the behavior of the series.-Relationships between datasets: Within the statistical functions, there is an option that allows you to examine the relationships between the variables in the series.-Highlight the recommended actions: The system uses visual clues in the graph to represent the actions taken, those about to be executed, and the corresponding indications.-Informative text boxes: Emergent messages.-Customization of geometric shapes: The graph uses nodes of different sizes to identify the elements of the workflow.
Direction:This grade of guidance focuses on providing alternatives and options for executing actions in the analysis process; guiding users through possible routes, strategies, or methods; and facilitating their selection based on their needs or context.
-Visual clues and priority signals: The user recognizes the system’s recommendation through visual cues (colors), accesses explainability through visualizations, and obtains additional information through text boxes.-Alternatives and options: The graph suggests different routes, actions, and algorithms for executing tasks and subtasks, as well as the possibility of interaction through entering or modifying parameters.
Prescription:The system automatically displays a list of algorithms for each task/subtask and presents a structured workflow sequence based on established rules.

#### 3.2.5. Guided Graph

We propose the design of a dynamic graph with a hierarchical structure. The hierarchy organizes nodes (tasks, subtasks, and algorithms) and edges that define their dependencies. The colors of nodes and edges adapt to user interactions, providing visual feedback on the workflow. The graph integrates guidance, explainability, and recommendations to suggest paths and nodes of interest. Its dynamic nature allows for analyzing evolution over time and facilitates fluid navigation between the overview and specific details. The structure of the graph is detailed below.

The graph is a diagram with three parent nodes representing tasks: Data Quality, Data Reduction, and Behavior Variables, connected by edges that indicate the dependencies between tasks.The traversal of the graph begins at the Start node and continues through the first level of nodes of the three tasks (parents), which are displayed larger than the other sublevels.Subtasks represent the next level, with nodes smaller than those for tasks and sometimes with another subtask level.At the last level are the nodes that represent the algorithms, which are smaller in size.The edges are colored based on the activities, as indicated in the legend, signifying a dynamic component based on the progress of the analysis process.The system connects the algorithm’s nodes to the next task’s nodes via borders, forming a user’s route.

### 3.3. Component Tool for Guided Visual Analysis

A visual guided tool, GUIVisWeb (https://github.com/flordeluz/VisWeb, URL accessed on 10 July 2025), was developed and implemented based on the conceptualization of guides carried out in the first component. It allows for the extraction and visualization of the characteristics of multivariate time series to perform preprocessing and analyze their behavior (Figure 7).

## 4. Experiments and Results

### 4.1. Proposal Configuration

The proposal was developed and implemented using Python version 3.12.8 and NodeJS 18.19.1 as a base on a GNU/Linux distribution with kernel 5.15.19 or higher. The hardware used for the experiments was a workstation with an AMD64 processor or $X86_{64}$ architecture, 16 GB RAM, 32GB swap space, 64 GB SSD, 128 GB recommended, and a GPU (optional).

The GUIVisWeb tool was implemented according to the specifications of a multi-layer design: the frontend was developed in Vue.js, the backend was developed in Python, and the data source layer consisted of CSV files.

### 4.2. Test Dataset

For testing, multivariate time series from different domains were selected, including extensive, raw air quality data from Brazil, Spain, and India. Climate data from four meteorological stations in the city of Arequipa, Peru, and a Bitcoin time series were also selected.

Spain Time Series: The time series dataset Air Quality in Madrid (2001–2018) was used, extracted from the Kaggle repository [46].Brazil Time Series: The Qualidade do Ar time series dataset was used, extracted from the São Paulo State Environmental Company, Brazil [47].India Time Series: The Air Quality Data in India (2015–2020) time series dataset was used, extracted from the Kaggle repository [48].Peru Time Series: The Arequipa time series dataset was used, extracted from the website of the National Meteorological and Hydrological Service of Peru—SENAMHI [49].Bitcoin Time Series: The BitCoin Historical Data dataset was used, obtained from https://www.investing.com/crypto/bitcoin/historical-data, URL (accessed on 10 July 2025).

### 4.3. Case Study

To carry out the tests, we validated the proposal using air quality data from Madrid (Spain), collected between 2001 and 2009 at station 28079001 (73,080 records). The datasets recorded data every hour; however, for visualization purposes, the system used sampling because of the large volume of data.

The process began with the visualization of a list of the raw time series (Madrid, India, Brazil, Arequipa, and Bitcoin). Upon selection, the stations were displayed, each with information on variables, null values, and the number of records. This gave the user an initial overview of the series (degree of orientation guidance) (Figure A4 in Appendix D).The visualization in Figure 7 is the primary interaction screen that shows the integrated workflow for preprocessing and analyzing the behavior of time series.The display of general information is complemented by more detailed information accessible via the Statistics and Time Series buttons. This availability provides information at the time it is needed (timeliness, explainability, transparency) regarding changes after preprocessing. It facilitates data interpretation throughout the analysis. The Statistics option displays information on the correlation matrix, bivariate analysis, and descriptive statistics (Figure A5 and Figure A7 in Appendix D). Time Series displays visual information about the series (Figure A8 in Appendix D).The graph guides the tasks to be performed through visual color cues, as indicated in the legend:
-Pending activities: Nodes and edges are displayed in red when the system recommends a route, task, subtask, or algorithm that has not been applied. The guide is generated from the analysis in the recommendation module, which evaluates the workflow.-Applied activities: Nodes and edges are displayed in blue to indicate actions that have been executed, providing a clear mental map of the process and facilitating workflow tracking.-Not pending activities: Nodes and edges are displayed in gray when actions cannot yet be executed due to pending tasks that must be completed first.-No action required: Nodes and edges are displayed in green when no further action is required.-Optional activity: Nodes and edges are displayed in orange when the decision to execute is optional and at the user’s discretion.
The system recommends treating null values first (red color), and, among the algorithms (7), the nodes rolling mean and random forest were recommended (displayed in darker shades). However, users can choose and execute a different algorithm than the one recommended (Figure A1 in Appendix A). This visual recommendation allows for a reduction in cognitive load as it simplifies decision-making. It focuses on interpreting results instead of comparing algorithms. Therefore, the recommendation is relevant, at the right time (opportunity), suggesting the best option (guide degree of direction).The explainability in Figure 4 shows six red boxes in the row of the random forest algorithm and one in the rolling mean. Of the two recommended algorithms, the one that performed best was random forest. The information text also indicates the recommended algorithm. Explainability through visualization and text clarifies the reasons for the recommendation (transparency and trust).After imputing null values, the system recommends running the Interquartile Range algorithm for outlier treatment (Figure A2 in Appendix B). The explanation of its recommendation shows that 100% of the outliers in the IQR are treated (Figure 5 about explainability of algorithm outliers).For the next task, the system optionally (orange) recommends the dimensionality reduction/PCA algorithm task (Figure A3 in Appendix C). As part of the explainability, a correlation matrix table is displayed to analyze the correlation between variables and decide which variable to reduce. KMO = 0.01; therefore, the PCA algorithm was recommended (Figure 6).The necessary preprocessing tasks were completed. The complete path of the actions performed is displayed in the graph (blue color). The series behavior nodes or buttons were enabled for their respective analysis (Figure A15 in Appendix E). Access through the button allowed us to select a specific range of the series (Figure A11 and Figure A12 in Appendix D). With access through the nodes, the behavior of the series was displayed for each variable (Figure A16 and Figure A17 in Appendix E).The system displays stacked windows on the left and right sides of the graph. They record executed actions that can be used to reverse the desired action.The tool features a set of buttons on the upper right corner of the main screen, enabling various actions and relevant information: HOME (select the time series to analyze), ASSETS (allows you to export the preprocessed series and import a new multivariate time series for analysis, NAVIGATION (primary screen), TIME SERIES (displays the series, trend, seasonality, spiral, and statistics). (See Appendix D and Figure A9, Figure A10, Figure A11, Figure A12 and Figure A14.)Once the analysis is complete, the user can export the data for future use, primarily for predictive models. This option is found in the ASSETS button.

Through the use case, we demonstrated the operation of the tool for guided visual analysis of the preprocessing and behavior of the series, in which all the conceptualization carried out in the first component was implemented: graph, workflow, guides, recommendations, and explainability.

### 4.4. Evaluation of the Explainability Model

This was carried out according to the latest publication by Musleh et al. [8], in which they propose a design model to build and present an explainable guide (GX). In our case, explainability is applied to help users understand the algorithm’s recommendations provided by the GUIVisWeb system.

To evaluate the explainability of our proposal, a survey was administered to 24 participants with little or no knowledge (novices). The survey consisted of 21 Likert-type questions (Q1–Q21) on a scale of 1 = “Strongly disagree” to 6 = “Strongly agree.” The questions evaluated the following characteristics:Clarity (Q1: Simplicity, Q2: Accuracy, Q3: Transparency)Timeliness (Q4: Selectivity, Q5: Efficiency)Interpretability (Q6: Completeness, Q7: Efficacy, Q8: Anthropomorphic)Feature Explanation (Q9: Relevance, Q10: Adaptability)Cognitive Relief (Q11: Efficiency, Q12: Persuasion)Confidence According to User Expectations (Q13: Competence, Q14: Consistency, Q15: Usability)Stages of the Trust Relationship (Q16: Initial, Q17: Intermediate, Q18: Final, Q19: Interactivity)Informed Decision-Making (Q20)Satisfaction (Q21)

Figure 8 shows the average of the questionnaire responses (questions Q1 to Q21) for the nine explainability evaluation criteria. These results demonstrate that the proposal received a positive rating for all criteria, with scores ranging from 5.37 to 5.65, particularly for the Timeliness and Cognitive Relief criteria. However, the Interpretability criterion has most significant possibility for improvement, particularly regarding the completeness and human nature of the explanations. These findings indicate a robust system that promotes user confidence in performing preprocessing tasks since the explanations meet expectations of clarity, timeliness, relevance, cognitive relief, interactivity, and satisfaction in each situation and stage of analysis.

### 4.5. Evaluation of the Guidance System

The evaluation was conducted based on the proposal made by Ceneda et al. [30]; considering that Ceneda is a pioneer in the guideline approach, it continues to work to improve and expand its impact in this area. Eight evaluation criteria were considered, as outlined in Section 2.5. The proposed guidelines were applied throughout the analysis process and are embodied in the GUIVisWeb application.

A survey was administered to 24 inexperienced participants. The survey consisted of 27 Likert-based questions (H1–H27) ranging from a scale of 1 = “Strongly disagree” to 7 = “Strongly agree”; eight criteria were evaluated:Flexibility (H1)Adaptability (H2: Guide change, H3: Experience adjustment)Visibility (H4: Easy identification, H5: Visible state and parameters)Controllability (H6: Explicit control, H7: Alternative suggestions, H8: Feedback, H9: Parameter adjustment)Explainability (H10: Understanding the results guided by the system, H11: Suggestions from the guides are easy to understand, H12: Understanding why the suggestions are provided, H13: Able to request explanations, H14: Able to trust the guides)Expressiveness (H15: Clear Language, H16: Unequivocal Coding)Timeliness (H17: Just-in-time guidance, H18: Does not disrupt workflow)Relevance (H19: The guide helps to overcome analysis dead-ends, H20: The guide helps to complete tasks by reducing errors, H21: The guide saves time, H22: The guide facilitates reasoning, H23: The guide helps answer questions about the data, H24: The guide is appropriate for the task being performed, H25: The guide helps make discoveries, H26: The guide helps generate new hypotheses about the data, H27: Applying the guide helps one to feel confident about the results.)

Figure 9 shows the averages of the questionnaire responses (questions H1 to H27) for the eight evaluation criteria of the proposed guideline. These results demonstrate that the proposal received a positive rating across all the guidance criteria, with scores ranging from 5.5 to 6.5.

## 5. Discussion and Future Work

### 5.1. Discussion on the Result of the Guidance System

The comparative analysis conducted in Table 1 of Section 2 shows that previous work focused on partial aspects. Theoretical approaches predominate over practical ones, more knowledge gaps are addressed than decision-making, and the cognitive aspect is prioritized over the technical. Furthermore, most work with guidance and direction guides, while the degree of prescription is little explored. Our proposal offers a comprehensive approach that covers these aspects:

Theoretical: It expands the conceptual model of guidance developed by Ceneda et al. [6], integrating interactive visual analysis with time series preprocessing, a critical phase usually left to the user.

Practical: It implements the GUIVisWeb tool, where a hierarchical dynamic graph organizes tasks, subtasks, and algorithms and combines navigation, recommendations, and explainability.

Knowledge: It provides support for minimally assisted preprocessing tasks, reducing uncertainty.

Decision: It not only presents alternatives but also recommends optimal algorithms for each task.

Cognitive: It incorporates explainability that facilitates understanding and trust in the recommendations.

Technical: It integrates specific preprocessing algorithms, expanding the scope beyond exploration.

Degrees of guidance: It encompasses orientation, direction, and prescription, strengthening decision-making assistance.

In addition, we conducted an empirical evaluation of the guide system through a survey administered to 24 participants, based on the criteria proposed by Ceneda et al. [6]. The results show positive ratings across all evaluated aspects, with averages between 5.5 and 6.5 on the Likert scale (1–7). Particularly noteworthy are the criteria of expressiveness, relevance, adaptability, visibility, and timeliness, reflecting its transparency, pertinence, flexibility, and clarity. However, controllability and explainability could be improved, particularly in the ability to request explanations for specific recommendations other than those generated by the algorithms. These findings reinforce the value of the proposal and point to concrete lines of future work.

### 5.2. Discussion of the Results of the Explainability of the Model

The survey results indicate that explainability was positively evaluated across all nine criteria proposed by Musleh et al. [8], with average scores above 5.3 on a 6-point Likert scale. The highest scores were obtained for the Timeliness and Cognitive Relief criteria, suggesting that users particularly valued the system’s ability to provide explanations promptly. These findings support the effectiveness of the proposed approach in improving the confidence and usability of novice users faced with complex preprocessing tasks. On the other hand, the relatively low score for Interpretability, especially regarding the completeness and anthropomorphic dimension of explanations, highlights an opportunity for improvement. Future improvements could focus on providing richer contextual details to strengthen user understanding further. Overall, the results demonstrate that the system successfully balances clarity, confidence, and decision support, thus fostering user engagement and confidence throughout the analysis process.

### 5.3. Discussion of the Results of the Case Study

The case study demonstrates the feasibility of the proposed approach, where the graph-based interface integrates recommendations, explainability, and workflow guidance to support the preprocessing of time series multivariate data, thereby reducing cognitive load and enabling informed decision-making. Although tasks, subtasks, and algorithms are prescriptively defined, the system allows for flexibility as the graph dynamically adapts when extending or reducing the workflow. The algorithms used, while not sophisticated, are widely employed in the state of the art and serve to demonstrate the guidance approach; future work could incorporate more advanced alternatives to enhance preprocessing capabilities further.

## 6. Conclusions

This work presents a new guidance-based approach to preprocessing multivariate time series that integrates recommendations, explainability, and interactive visual analysis, embodied in the GUIAVisWeb tool. The preprocessing phase is explicitly incorporated into interactive visual analysis through a systematic workflow of tasks, subtasks, and algorithms that facilitates reproducibility and reduces subjectivity at this critical stage. A modular architecture was designed and validated, featuring components for configuration, task flow, recommendations, guidance degree, and graph representation, enabling flexibility to extend or adapt the workflow.

A significant contribution is the introduction of explainable guidance (XG), which combines algorithmic recommendations with visual and textual justifications, marking the first integration of explainability into a guidance system. Empirical evaluation with users confirmed the effectiveness of the approach, yielding high scores in usability (5.56/6), explainability (5.56/6), and guidance quality (6.19/7). These results demonstrate that the system reduces cognitive load, enhances user understanding, and supports informed decision-making, consolidating the contribution as a significant advance in the state of the art.

## Figures and Tables

**Figure 1 sensors-25-05617-f001:**
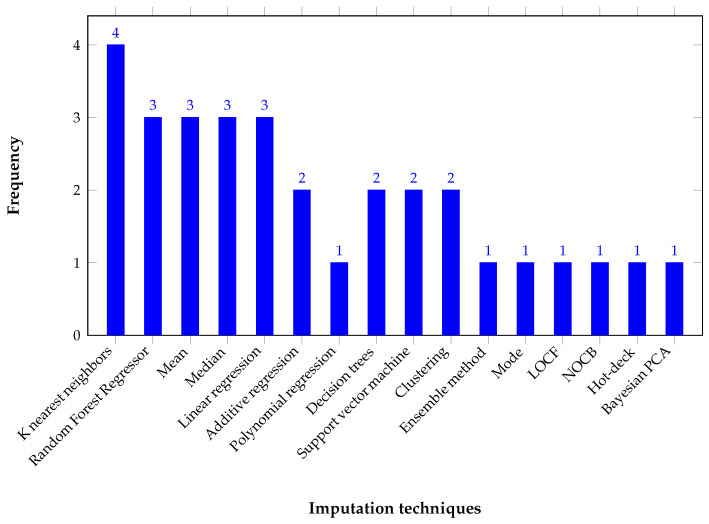
Frequency of imputation techniques according to the review.

**Figure 2 sensors-25-05617-f002:**
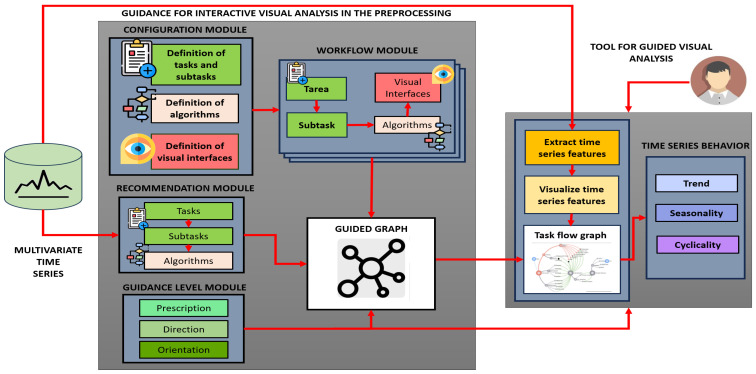
Architecture of the guidance for interactive visual analysis in multivariate time series preprocessing.

**Figure 3 sensors-25-05617-f003:**
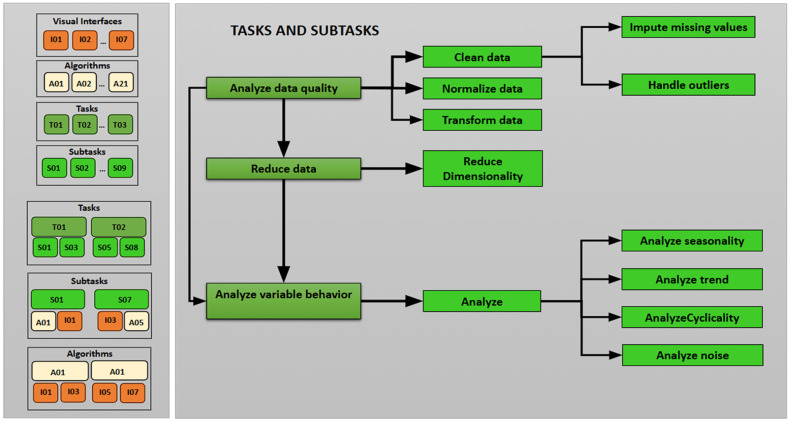
Task and subtask workflow.

**Figure 4 sensors-25-05617-f004:**
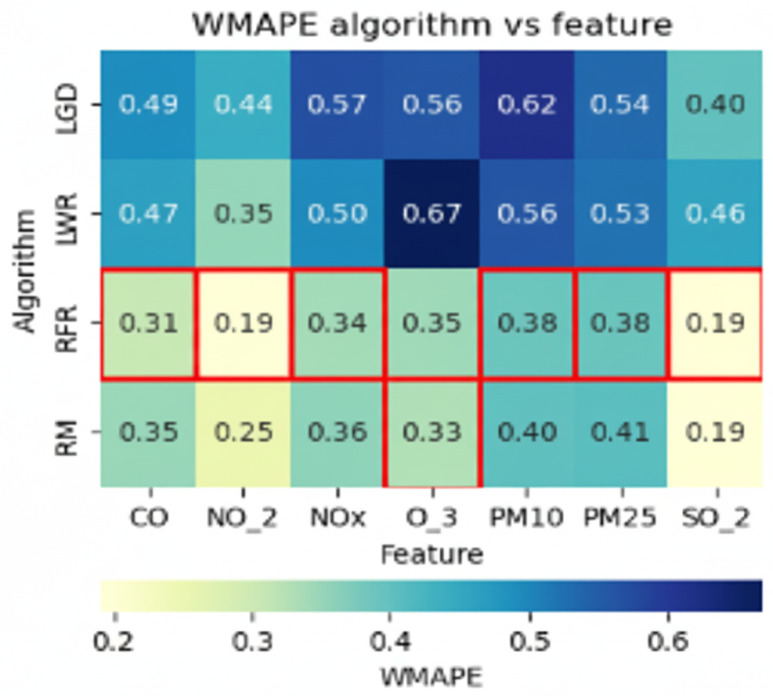
Explainability: WMAPE measure by algorithm and variable.

**Figure 5 sensors-25-05617-f005:**
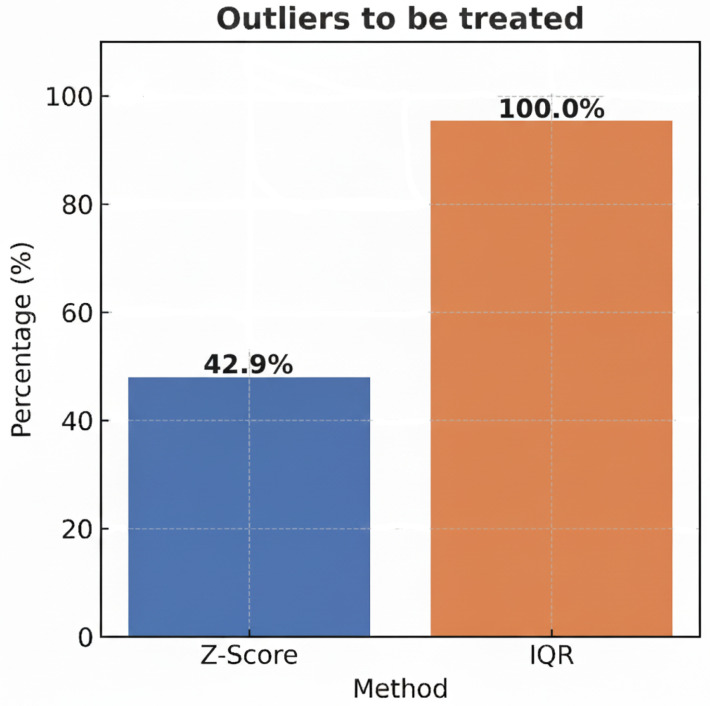
Explainability: percentage of outliers treated.

**Figure 6 sensors-25-05617-f006:**
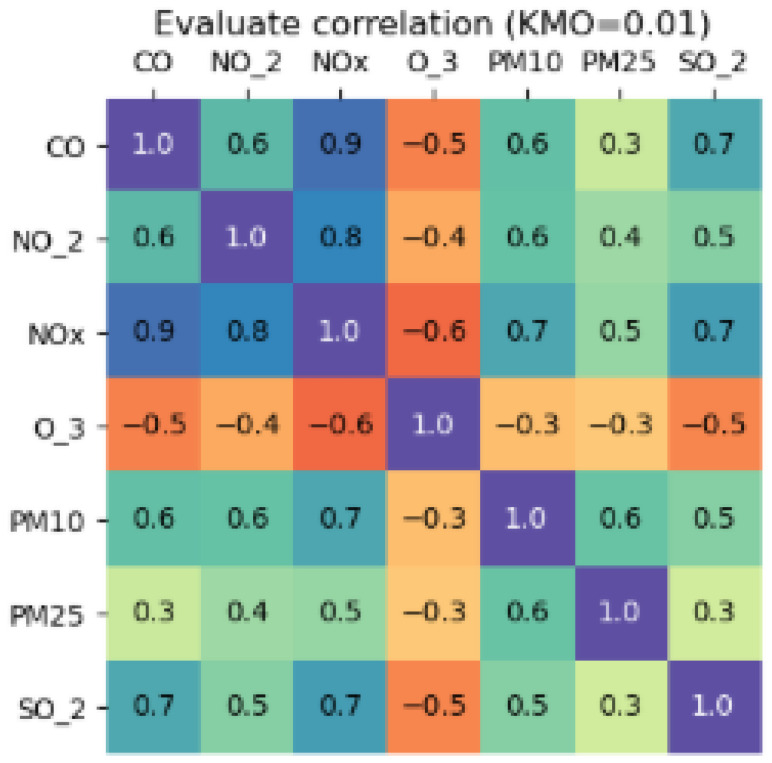
Explainability: evaluating correlation of variables with KMO.

**Figure 7 sensors-25-05617-f007:**
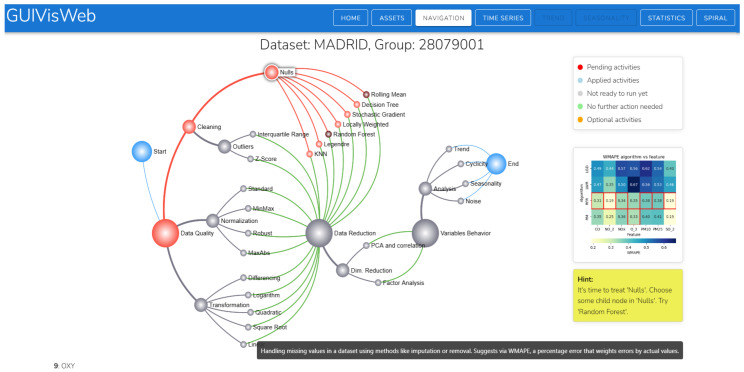
Tool for guided visual analysis.

**Figure 8 sensors-25-05617-f008:**
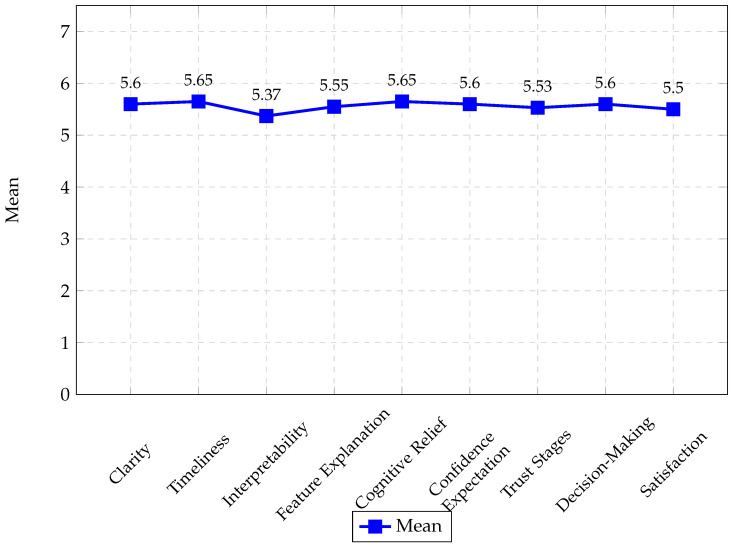
Comparison of the average means of the criteria that evaluated the proposed explainability, based on questions Q1 to Q21.

**Figure 9 sensors-25-05617-f009:**
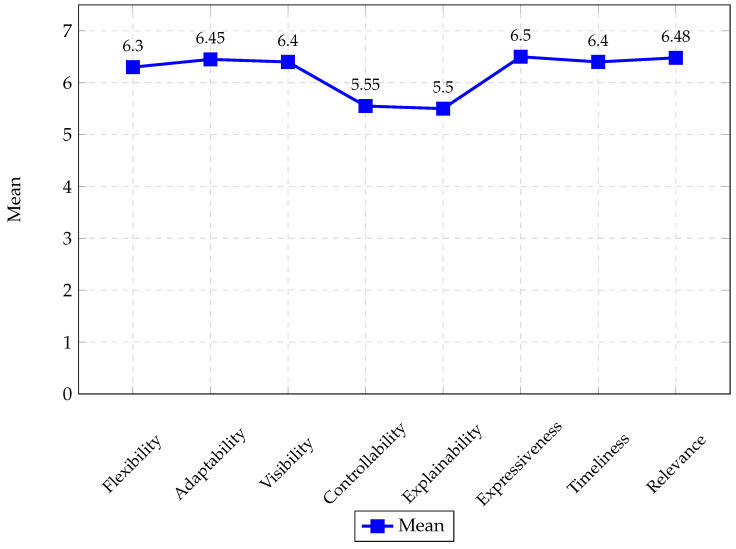
Comparison of the average means of the 8 criteria that evaluated the proposed guides, based on questions H1 to H27.

**Table 1 sensors-25-05617-t001:** References to guide works in visual analysis. (1) Theoretical, (2) Practical, (3) Knowledge gap, (4) Decision-making, (5) Technical aspect, (6) Cognitive aspect, (7) Direction, (8) Orientation, (9) Prescriptive.

Reference	(1)	(2)	(3)	(4)	(5)	(6)	(7)	(8)	(9)	Characteristics
Ceneda et al. [6]	X		X			X	X	X	X	(1) Conceptual model of guidance in VA (characterizing). (3) Importance of perception and cognition in obtaining knowledge. (7), (8), (9) Author of the three degrees of guidance.
Ceneda et al. [14]		X	X		X			X		(2) Design and implementation of an interactive system using a spiral visualization. (3) Difficulty for users to configure cycle length parameters. (5) Implementation and design of the interface. (8) Visual cues on the spiral suggest cycle length configurations based on statistical results.
Ceneda et al. [2]	X		X			X		X		(1) Approach to a conceptual framework for guide designers in VA. (3) Lack of understanding of the data. (6) The guide reduces mental load by guiding users through the initial stages of analysis. (8) The framework suggests directions and steps for designers, using cues such as requirements and examples to guide the process without imposing complete solutions.
Palomino et al. [21]	X	X	X	X	X	X	X	X		(1) Proposes guidance and describes interfaces and operators for time series. (2) Implements a task flow tool. (3) Helps with preprocessing tasks. (4) Suggests alternatives of algorithms and routes to follow. (5) Shows algorithms that can be run on preprocessing tasks. (6) Gaining knowledge. (7) Alternatives to algorithms and tasks. (8) Visual signs of the routes to follow.
Luboschik et al. [16]	X		X			X	X			(1) Conceptual framework. It seeks to help users understand patterns in multiscale data through visual guidance. (3) Identifies interesting regions in multiscale data. (6) Focuses on perceptual guidance and visual design. (7) Provides heterogeneity metrics to guide the user toward regions of interest.
May et al. [17]	X		X			X	X			(1) Proposes navigation concepts and methods, with a focus on conceptual frameworks. (3) Seeks to facilitate the exploration and understanding of large graphs through visual techniques. (6) Focuses on perceptual and visual design aspects. (7) Provides alternatives to guide navigation.
Streit et al. [3]	X		X			X	X			(1) Proposes a model-driven design process and describes interfaces, operators, and data. (3) Facilitates the understanding and visual analysis of heterogeneous data through a structured design. (6) Focuses on aspects of visual design and perceptual analysis. (7) Presents alternatives through defined interfaces and routes.
Gladisch et al. [18]	X		X			X	X			(1) Based on the degree of interest (DOI), it describes visualizations and algorithms, focusing on concepts and a test implementation without providing extensive practical details. (3) Seeks to facilitate the visual exploration of hierarchical graphs through recommendations. (6) Focuses on visual perception and user assistance with visual cues. (7) Provides recommendations through visual cues and DOIs, guiding the user to nodes of interest.
Palomino et al. [15]		X	X		X			X		(2) Implements a guided tool to find cyclical patterns. (3) Setting parameters, especially the cycle length. (5) Spiral implementation to detect cyclic patterns and star-like glyphs. (8) Visual signals on the cycle length parameter.
Collins et al. [7]	X		X			X	X	X	X	(1) Proposes a conceptual framework, discussing abstract concepts such as types of guidance, objectives, and requirements. (3) The guide helps analysts understand unknown data, identify patterns, and build mental models. (6) Pattern extraction, bias mitigation, and cognitive load reduction. (7), (8), (9) The framework recommends using all three degrees of guidance.
Han et al. [19]	X			X	X	X	X	X		(1) Presents a conceptual framework for designing guides in visual analytics (VA) based on decision points and decision support (MCDA), with emphasis on the design and implementation stages, but includes an initial prototype as a practical example. (4) Defines guidance as support at decision points using MCDA to evaluate alternatives. (5) Presents a cognitive bias because it focuses on reducing cognitive load, mitigating biases, and adapting the guide to the user’s specific needs. (6) Includes technical aspects such as integration with GIS and MCDA. (7) Alternatives are evaluated using multicriteria decision analysis (MCDA). (8) Provides an overview by identifying decision points.
Perez et al. [20]	X		X	X	X	X	X	X	X	(1) Proposes a model, a typology, and a conceptual analysis of guidance tasks in mixed visual analysis environments. (3) Focuses on supporting data exploration and understanding. (4) Considers decision-making in analytical contexts. (5) Adapts user–system interactions. (6) Task decomposition and guidance systems. (7), (8), (9) The model considers the three degrees of guidance.

**Table 2 sensors-25-05617-t002:** Tasks identified in the references. (1) Transformation, (2) Integration, (3) Normalization, (4) Missing data imputation, (5) Noise, (6) Outliers, (7) Dimensionality reduction, (8) Data partitioning, (9) Data aggregation.

Reference	(1)	(2)	(3)	(4)	(5)	(6)	(7)	(8)	(9)	Total
Garcia et al. [22]	X	X	X	X	X					5
Fan et al. [5]	X		X	X		X	X	X		6
Maharana et al. [24]	X		X	X	X		X			5
Ccetin et al. [23]	X		X	X	X		X		X	6
Mallikharjuna et al. [25]			X	X		X	X			4
Palomino et al. [21]	X		X	X		X	X			5
Total	5	1	6	6	3	3	5	1	1	31

**Table 3 sensors-25-05617-t003:** Techniques identified for the imputation of null values.

Imputation Technique	Emmanuel et al. [26], 2021	Joel et al. [28], 2022	Thomas et al. [29], 2021	Hasan et al. [27], 2021	Total
K-nearest neighbor (KNN)	X	X	X	X	4
Random forest regressor	X		X	X	3
Mean	X	X		X	3
Median	X	X		X	3
Linear regression	X	X		X	3
Additive regression	X			X	2
Polynomial regression				X	1
Decision trees	X	X			2
Support vector machine (SVM)	X			X	2
Clustering	X		X		2
Ensemble method	X				1
Mode		X			1
LOCF (Last Observation Carried Forward)		X			1
NOCB (Next Observation Carried Backward)		X			1
Hot-deck		X			1
Bayesian PCA (Bayesian principal component analysis)				X	1

**Table 4 sensors-25-05617-t004:** Task definition.

Group	Task	Code
Preprocessing	Data Quality	T01
	Data Reduction	T02
Time Series Behavior	Time Series Behavior	T03

**Table 5 sensors-25-05617-t005:** Subtask definition.

Subtask	Code
Null Values	S01
Outliers	S02
Normalization	S03
Transformation	S04
Data Reduction	S05
Trend	S06
Seasonality	S07
Cycle	S08
Noise	S09

**Table 6 sensors-25-05617-t006:** Definition of algorithms.

Algorithms	Code
Rolling Mean and Moving Median	A01
Decision Tree	A02
Stochastic Gradient Boosting	A03
Robust Locally Weighted Regression	A04
Random Forest Regressor	A05
Legendre Polynomials	A06
K Nearest Neighbor with Bagging Improvement (KNN)	A07
Interquartile Range (IQR)	A08
Scoring Method (Z-score)	A09
StandardScaler	A10
MinMaxScaler	A11
Robust-Scale	A12
Differencing	A13
Logarithm	A14
Quadratic Transformation	A15
Square Root Transformation	A16
Linear Transformation	A17
Principal Component Analysis (PCA) and Correlation	A18
Factor Analysis	A19
Seasonal Decompose	A20
Fourier Transform	A21

**Table 7 sensors-25-05617-t007:** Assigning visual interfaces to tasks/subtasks.

Visual Interfaces	Information
I01 ScatterPlot	Statistics: Bivariate analysis
I02 Lineplot	Time series, behavior
I03 Spiral	Univariate cyclicality
I04 BoxPlot	Statistics: Data distribution
I05 Histogram	Statistics: Bivariate analysis,
	data distribution
I06 Heatmap	Explainability of null values, outliers
I07 Star Glyph	Multivariate relationship
I08 Dynamic graph with hierarchical structure	Workflow
I09 Bar Graph	Normalization, transformation
I10 Correlation matrix	Statistics, reduction of
	dimensionality

**Table 8 sensors-25-05617-t008:** Hierarchical workflow organization.

Group	Task	Subtask	Algorithms
G1	T01 Data Quality	S01 Null values	A01 Rolling Mean and Moving Median
			A02 Decision Tree
			A03 Stochastic Gradient Boosting
			A04 Robust Locally Weighted Regression
			A05 Random Forest Regressor
			A06 Legendre Polynomials
			A07 Bagged k-Nearest Neighbor (KNN)
		S02 Outliers	A08 Interquartile Range (IQR)
			A09 Scoring Method (Z-score)
		S03 Normalization	A10 StandardScaler
			A11 MinMaxScaler
			A12 MaxAbsScaler
			A13 Robust-scale
		S04 Transformation	A15 Differencing
			A16 Logarithm
			A17 Quadratic Transformation
			A18 Square Root Transformation
			A19 Linear Transformation
	T02 Data Reduction	S05 Dim.Reduction	A20 Principal Component Analysis (PCA)
			A21 Factor Analysis
G2	T03 Data Behavior	Trend	
		Cyclicality	A22 Fourier Transform
		Seasonality	
		Noise	Seasonal Decompose

**Table 9 sensors-25-05617-t009:** Representative Response: The tables should be displayed separately, as they were previously. example of data with null values.

0	1	2	3	4	5	6	7	8	9	10	11	12	13	14	15
3	2	5	6	NaN	2	1	NaN	4	6	NaN	NaN	9	NaN	4	NaN

**Table 10 sensors-25-05617-t010:** Generating Two Datasets: Complete Data and Null Data.

Complete Data											Null Data					
**0**	**1**	**2**	**3**	**5**	**6**	**8**	**9**	**12**	**14**		**15**	**4**	**7**	**10**	**11**	**13**
3	2	5	6	2	1	4	6	9	4		NaN	NaN	NaN	NaN	NaN	NaN

**Table 11 sensors-25-05617-t011:** Generación de tres datasets.

0	1	2	3	5	6	8	9		12	14		15	4	7	10	11	13
3	2	5	6	2	1	4	6		9	4		NaN	NaN	NaN	NaN	NaN	NaN

## Data Availability

The data presented in this study are available in Air Quality Dataset at urlhttps://www.kaggle.com/datasets/decide-soluciones/air-quality-madrid (accessed on 10 July 2025). Qualidade do Ar time series dataset at https://sistemasinter.cetesb.sp.gov.br/ar/php/mapa_qualidade_rmsp.php (accessed on 10 July 2025). Air Quality Data in India at https://www.kaggle.com/datasets/rohanrao/air-quality-data-in-india (accessed on 10 July 2025). National Meteorological and Hydrological Service of Peru—SENAMHI at https://www.senamhi.gob.pe/site/descarga-datos/ (accessed on 10 July 2025). BitCoin Historical Data dataset at https://www.investing.com/crypto/bitcoin/historical-data (accessed on 10 July 2025).

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
