# Peer review of "Guidance for Interactive Visual Analysis in Multivariate Time Series Preprocessing"

_sensors, 2025, doi:10.3390/s25185617_

Round 1
Reviewer 1 Report
Comments and Suggestions for Authors
The paper could be published, if it is presented differently by mainly improving the clarity, focus and brevity. Please consider the following revisions. The presentation would benefit from moving some material into Supplementary file, or to Appendices. In general, it is much more valuable to have a short and well presented paper with clear contributions, than long paper, which is unclear and its contributions cannot be easily identified.
- The paper title: my understanding is that the paper reports a general framework for (pre-) processing and especially visualizing multivariate time series data.
- The main objective seems to be to configure and choose the right pre-processing. This needs to be already clear from Abstract. The abstract should define the problem/objective (1/3 of the text), how that problem was solved (1/3), and what has been achieved (remaining 1/3).
- Add a short paragraph to the end of Introduction explaining the paper structure.
- In Related Work, Table 2 appears to be continuation of Table 1, so the former should be labeled as Table 1 (continuation). In Table 3 and 4, why not to use 'x' as in Table 1 instead of 0/1?
- Section 3 title: Proposed Framework. Figure 5 would look nicer with white background boxes. Many tables from this section could be moved to appendices or supplementary to improve clarity and brevity. The main focus should be on describing and explaining the methods of the proposed framework, leaving unnecessary details to appendices. Figure 5-8 should be better drawn, or completely removed unless they are really important to explain the methods. Figure 9,11,13,16 seems to be repeated as much nicer Figure 15, so the former are redundant. In general, please avoid using screenshots in the main text, or move the screenshots to supplementary (not even to appendices).
- Section 3 should be divided into Section 3 Methods and Section 4 Results. The former explains the proposed frameworks, and key steps. The latter should demonstrate how the framework is used for different type of time series datasets including how to configure pre-processing.
- The paper would benefit from having Discussion section to evaluate what has been achieved, how the proposed framework is better than other methods commonly used (e.g. in the literature), identify any shortcomings and limitations, and also outline the future work.
Author Response
Dear reviewer.
Our manuscript has been updated in every section, and all your comments have been answered. Some may not meet all your expectations, but we ask for your patience and understanding that this work is not intended to be perfect or a top-notch proposal.
Thank you for your comments. We appreciate your time and interest and hope to have your acceptance for publication. You will find the responses to your comments in the attached PDF file.
Best regards.
Flor de Luz Palomino Valdivia.

Reviewer 2 Report
Comments and Suggestions for Authors Referee report for the manuscript: This paper addresses the challenges associated with the preprocessing phase in multivariate time series analysis using interactive visual analytics (IVA). It highlights that analyzing such time series is complex due to characteristics such as variability, scalability, and multivariance, which require specialized methods to support users in making informed technical decisions. The paper presents a comprehensive framework that integrates automated guidance with user cognitive needs, focusing on the often-overlooked preprocessing phase in current IVA systems. The proposed framework consists of six main components and was evaluated using air quality data from Brazil, India, and Madrid, as well as meteorological data from Peru. The results demonstrate the system's effectiveness in terms of usability, comprehensiveness of the analysis process, and quality of outcomes. Comments Despite the quality of the paper, there are some points that could be improved: 1.The text is rich in detail; however, some sentences are overly long and complex, which can make them difficult to follow. It is advisable to break these into shorter, more direct sentences to enhance readability. 2.While the introduction provides extensive background information, the uniqueness of your research should be emphasized in one or two concise sentences early in the text. This will immediately inform the reader of what is new and innovative about your work. 3.Certain word choices can be improved: The term "pistas" in line 169 appears to be an untranslated Spanish word; it should be replaced with "hints" or "clues". In some cases, "address the question" can be replaced with "answer the question" to avoid redundancy. The verb "utilize" can often be simplified to "use" in less formal contexts or where clarity is a priority. 4.Some paragraphs contain repeated phrases; reducing such repetition will improve flow and conciseness. 5.Adding a quantitative comparison with other systems would further strengthen the paper by clearly demonstrating the superiority and effectiveness of the proposed framework. If the above points are adequately addressed, the paper would be suitable for publication.Author Response
Dear reviewer.
Our manuscript has been updated in every section, and all your comments have been answered. Some may not meet all your expectations, but we ask for your patience and understanding that this work is not intended to be perfect or a top-notch proposal.
Thank you for your comments. We appreciate your time and interest and hope to have your acceptance for publication. You will find the responses to your comments in the attached PDF file.
Best regards.
Flor de Luz Palomino Valdivia.

Reviewer 3 Report
Comments and Suggestions for Authors
In this manuscript, the authors introduce a guidance framework for Interactive Visual Analytics in the preprocessing of multivariate time series. This framework comprises several components: (1) the automated recommendation of tasks, subtasks, and algorithms; (2) the workflow structure; (3) the application of guidance degrees, specifically orientation, direction, and prescription; (4) the utilization of a dynamic graph with a hierarchical structure for the workflow; (5) the development of an interactive visual tool; and (6) the explainability of the recommended algorithms. The proposal was qualitatively evaluated using air quality datasets from Brazil, India, and Madrid, as well as meteorological data from Peru, focusing on the model’s explainability and the guidance system’s effectiveness. The authors aim to address the complexity of this task by creating a system that offers automated recommendations, workflow management, and an explainable model. The motivation is well-founded because guiding users through complex analytical preprocessing is a significant challenge in data science and visual analytics.
However, this work is undermined by a lack of perceived novelty, substantial presentational issues, and a general sense of superficiality that diminishes its research contribution. The authors must revise the manuscript to address these core problems.
1. The primary weakness of this manuscript is its failure to establish a clear and novel scientific contribution to the field. The core proposal is the development of an interactive visual tool that guides the users. While the integration of features such as automated recommendations, workflow structures, and explainability is a valid engineering effort, this manuscript does not convincingly argue that this combination represents a fundamental advancement in the literature.
2. The field of visual analytics is replete with tools that aim to guide users through complex data-analysis pipelines. The authors must explicitly address the question: What is the unique scientific or theoretical contribution of this work beyond the implementation of a new tool? Is this a new guidance model? Is there a novel recommendation algorithm? A demonstrable leap in usability over existing systems Without a clear answer, the work appears to be an application of established concepts rather than foundational research.
The quality of the manuscript raises serious concerns regarding the rigor of the study.
Excessive Length: The paper is excessively long. A lengthy and unfocused manuscript often obscures the key message and suggests that the authors have not effectively distilled their primary contributions to the field. The authors should significantly shorten the paper, focusing on the most critical aspects of their methodology and the results.
Poor Visualizations: All visuals should be revised to ensure legibility and comprehensibility.
Language and Professionalism: Untranslated Spanish phrases in English-language manuscripts are unacceptable. This points to a lack of thorough proofreading and general carelessness in preparation that erodes confidence in the research itself.
Author Response

(The authors gave the same response as above.)

Round 2
Reviewer 1 Report
Comments and Suggestions for Authors
The authors mostly address my comments and suggestions. However, I still noticed a few areas for improvement. Please consider the following further revisions.
- The paper title could be improved; for example: A Guiding Tool for Interactive Visual Analysis and Pre-Processing of Multivariate Time Series
- The URL to the developed webtool could be added to the Abstract, if it is publicly accessible.
- The paper structure on l. 69-73 belongs to the very end of first section.
- Figure 4 is blurrer.
- The number of sub-(sub)-sections is confusing. E.g. on p. 20, there is sub-section 6, which is followed by sub-section 3.2.4.
- Conclusions should be 1-2 paragraphs of text, not a bullet point list.
- In appendices, please add text to explain what is in the figures, possibly with references ot the main text where these figures are discussed in more detail.
Author Response
Thank you very much for taking the time to review this manuscript.
Please find the detailed responses below and the corresponding revisions/corrections highlighted/in track changes in the re-submitted files.

Reviewer 3 Report
Comments and Suggestions for Authors
The paper has been reviewed in a proper manner.
Author Response

(The authors gave the same response as above.)
